# Genomic Epidemiology of Vancomycin-Resistant *Enterococcus faecium* Isolates with Full and Truncated *vanA* Operons from Russian Hospitals

**DOI:** 10.3390/antibiotics14090858

**Published:** 2025-08-25

**Authors:** Anna Slavokhotova, Andrey Shelenkov, Yulia Mikhaylova, Lyudmila Petrova, Vitaly Gusarov, Mikhail Zamyatin, Vasiliy Akimkin

**Affiliations:** 1Central Research Institute of Epidemiology, Novogireevskaya Str., 3a, 111123 Moscow, Russia; slavokhotova@cmd.su (A.S.); mihailova@cmd.su (Y.M.); akimkin@pcr.ms (V.A.); 2National Medical and Surgical Center Named After N.I. Pirogov, Nizhnyaya Pervomayskaya Str., 70, 105203 Moscow, Russia

**Keywords:** antimicrobial resistance, healthcare-associated infections, MLST typing, truncated *vanA* operon, virulence factors, whole genome sequencing

## Abstract

**Background**: Vancomycin-resistant *Enterococcus faecium* (VREfm), particularly *vanA*-positive strains, represents a growing threat in hospital settings worldwide. These bacteria are able to survive under severe environmental conditions, including high temperatures and saline concentrations. High genome plasticity and advanced ability of inheriting antimicrobial resistance determinants defined the success of *E. faecium* as a hospital pathogen. **Methods**: This study presents the whole genomic characterization of *vanA*-positive VREfm isolates, analyzing 10 clinical isolates collected from three tertiary care hospitals in Moscow, Russia. Several typing approaches, including two MLST schemes and cgMLST profiles, were used to elucidate the relationship between the isolates. Phylogenetic analysis placed the isolates in context with global VREfm populations, demonstrating both local clonal expansion and possible international connections. Phenotypic and genomic antimicrobial resistance profiles were obtained, as well as data regarding the repertoire of virulence factors and plasmid content. **Results**: Whole genome sequencing revealed that all isolates belonged to the clinically significant CC17 lineage, specifically sequence types ST80 and ST552. Notably, two isolates possessed truncated Tn1546-type transposons lacking *vanY* and *vanZ* genes, representing a potentially emerging variant of the *vanA* operon in Russian clinical settings. A plasmid carrying a truncated *vanA* operon was reconstructed using long-read sequencing. **Conclusions**: The study highlights the utility of genomic investigation for tracking resistance mechanisms and strain dissemination, providing crucial baseline data for epidemiological surveillance of infections caused by VREfm in Russia. These findings emphasize the need for continued genomic monitoring to understand the evolution and spread of antimicrobial resistance in clinically important enterococcal lineages.

## 1. Introduction

Enterococci are facultative anaerobic Gram-positive bacteria belonging to the *Enterococcaceae* family [1]. These bacteria are able to survive under severe environmental conditions, including high temperatures (60 °C for 30 min) and saline concentrations (6.5% NaCl at pH 9.6) [2]. Enterococci can be used for both the fermentation of some meat and cheese products and as natural probiotics [3]. Although enterococci are present in the gastrointestinal (GI) tract of humans and animals as natural colonizers, in some cases, they can cause urinary tract infections, endocarditis, meningitis, soft-tissue and neonatal infections, bacteremia, sepsis, and other complications [3,4].

Two major representatives, *Enterococcus faecalis* and *E. faecium*, are of particular clinical interest. The first one is the cause of both hospital-acquired (HA) and community-associated (CA) infectious diseases, including various human mouth infections, while the second is usually responsible for HA diseases only [4,5]. While *E. faecalis* possesses higher virulence, *E. faecium* induces the diseases with higher mortality rates (up to 60–70%) [4,5] due to several special features, including a high genome plasticity and a high ability to acquire and inherit antimicrobial resistance (AMR) determinants [6]. These properties contribute to the emergence of multidrug-resistant (MDR) isolates possessing resistance to various classes of antibiotics. Since the 1960s, strains with resistance to streptogramines, aminoglycosides, macrolides, lincosamides, tetracyclines, glycopeptides, and cyclic lipopeptides have been discovered [7,8]. From the 1990s, a number of vancomycin-resistant isolates of *E. faecium* (VREfm) increased drastically; moreover, some of them gave rise to hypermutable isolates [9] that were extremely dangerous for immunocompromised patients and persons undergoing prolonged hospitalization [10]. Finally, in 2017, the WHO placed VREfm into a high-priority list of pathogens for research and development of new antibiotics, and this listing was confirmed in 2024 [11].

Among nine discovered phenotypes of vancomycin-resistant *Enterococci* (VRE), six (A, B, D, G, M, and N) were found in *E. faecium*. It should be noted that A-type of *E. faecium* is referred to as a major agent of nosocomial VRE infections because of its high resistance to glycopeptide antibiotics [12]. The isolates of this phenotype, together with D- and M-type of *E. faecium,* also exhibit resistance to teicoplanin. VRE phenotypes are encoded by the resistance gene cluster, which consists of 5 to 7 genes, three of which (*vanH*, *A*, *X*) are indispensable for resistance, while two regulatory genes (*vanR*, *S*) are also required, but they can change their location within the operon [13]. Additional genes (*vanY*, *Z*) are not essential for vancomycin resistance; however, *vanZ* participates in teicoplanin resistance and became a part of several operons. Most vancomycin resistance (VR) genes have inducible expression and can be transferred by conjugation, although the D-type of *E. faecium* has chromosomally-located VR genes.

It should be noted that molecular epidemiological analysis widely uses MLST for the characterization of VRE isolates. This analysis is based on allelic differences of housekeeping genes located in the core genome, while the information regarding mobile genetic elements and genomic relationships is not reflected. In addition, because of the limited information provided by MLST, it is impossible to trace recombination between isolates, as well as to properly determine ancestry relationships [7]. In contrast, whole genome sequencing (WGS) allows us to distinguish isolates with more accuracy and sensitivity, to consider recombination events and acquisition of foreign mobile elements [3].

In this study, we performed an integrated phenotypic and WGS analysis of ten VREfm isolates collected from three tertiary-care hospitals in Moscow, Russia. Comprehensive molecular typing was conducted to determine isolate relatedness, along with in-depth characterization of AMR determinants and virulence-associated genes. Additionally, plasmid content was analyzed, with three isolates subjected to long-read sequencing to achieve high-confidence plasmid assembly and reconstruction. Two isolates with a reduced *vanA* operon lacking *vanY* and *vanZ* were revealed. The data provided will contribute to epidemiological studies of *E. faecium* and provide additional insights into vancomycin resistance acquisition and dissemination within the population of this important pathogen.

## 2. Results

### 2.1. Isolate Typing and Comparison with Reference

The metadata for the ten isolates studied in combination with MLST typing results are provided in Table 1.

According to the older MLST scheme by Homan et al. (MLST^Hom^) [14], all isolates were assigned to CC17^Hom^, and eight of them belonged to ST80^Hom^. The newer scheme proposed by Bezdicek et al. (MLST^Bez^) [15] provided more subtle isolate differentiation with three CCs and five STs, which better reflected the sample diversity, as will be discussed below.

CriePt1877 possessed a novel *narB* allele (a part of MLST^Bez^ scheme), which was uploaded to the PubMLST database and assigned number 201, and thus this isolate was also assigned a novel ST^Bez^ denoted as ST1523^Bez^. Although MLST-based clustering can provide some useful epidemiological insights, cgMLST gives much higher resolution. The minimum spanning tree based on cgMLST profiles for the isolates under study is shown in Figure 1.

The cgMLST profiles for the ten isolates reported by us and reference isolates discussed in the manuscript are given in Appendix A.

Based on the cgMLST analysis, it is easy to see that MLST^Bez^ better reflects the relationship between the isolates since it assigns CriePt972 and CriePt1087, which are rather distant from CriePir271 and 272, to different ST^Bez^ (145 vs. 144). At the same time, CriePir1878 and CriePir1879, which were within 18 cgMLST alleles from CriePt1087, were assigned to a different ST^Bez^ than CriePt1087, while CriePt972, which was almost 300 alleles apart from it, was assigned to the same ST^Bez^ as the latter.

We also used cgMLST profiles to compare our isolates with those from Genbank with the same STs. Most reference isolates differed from their CriePir and CriePt counterparts by more than 200 alleles, and thus were not of interest for further comparison. However, the group of four isolates with 16–20 allele differences to CriePir337 was revealed, which contained SCPM-O-B-8940, SCPM-O-B-8941, SCPM-O-B-8942, and SCPM-O-B-8944 strains isolated from a human digestive system in Russia in 2020, and which also carried *vanA* operons.

Another relatively similar reference isolate was NIZ171 (Genbank acc. GCA_017639605.1) obtained in Malaysia in 2019, which had 78 allele differences to CriePt1087. However, this number of differences indicated a relatively large divergence, and NIZ171 did not carry any *van* genes.

### 2.2. Phenotypic and Genomic AMR and VanA Operon Structures

All isolates were resistant to ampicillin, gentamicin, levofloxacin, teicoplanin, and vancomycin, but susceptible to linezolid and tigecycline. There was good coherence between the phenotypes and genomic determinants of AMR, which can be seen in Figure 2.

Ampicillin resistance was predicted to be associated with multiple point mutations in the *pbp5* gene (V24A, S27G, and others, see Appendix A for a full list of these mutations) reported earlier [16], and levofloxacin resistance was attributed to point mutations in *gyrA* (S83I) and *parC* (S80I) genes. The isolates were also supposed to be resistant to macrolides, in particular, to erythromycin, due to the presence of *msrC* and *ermB/ermT* genes, but this antibiotic was not included in the panel tested since no cutoff value was provided for it in EUCAST guidelines. In addition, *cat* family genes providing resistance to chloramphenicol were found in CriePir271–273, but this antibiotic was not tested either due to the same reason. Tetracycline resistance genes *tet(M)* and *tet(L)* were revealed in five CriePir isolates. Certain mutations in *tet(L)* genes were also found to increase tigecycline resistance [17], but such mutations have not been revealed in these isolates. In addition, no isolates contained *tet(X)* genes known to provide tigecycline resistance [18], and all isolates were susceptible to this drug according to the phenotypic resistance analyses. Transferrable linezolid resistance genes or 23S rRNA gene mutations known to confer such resistance were not revealed in any of the isolates, which corresponds to the phenotypic susceptibility. However, the most interesting point was the resistance to vancomycin and teicoplanin conferred by the *vanHAX* operon, which will be described in more detail.

Complete assembly and annotation of vancomycin resistance operons revealed conserved *vanA*-type gene clusters across all isolates (100% sequence identity to reference *vanA* operons). Comparative genomic analysis demonstrated the consistent chromosomal integration of these resistance determinants within Tn1546-type transposons, confirming their classification as VanA phenotype VREfm strains. The preservation of canonical Tn1546 structural elements (including intact transposition genes) suggests maintained mobility potential among the studied isolates. Eight isolates of *E. faecium* contained a full-length *vanA* operon comprising seven genes (*vanRSHAXYZ*). The isolates CriePir337 and CriePir1877 were lacking both accessory genes (*vanY*, *vanZ*) and consisted of only five genes (*vanRSHAX*).

The operon structure for all isolates was very similar to the one presented by the Tn1546.1_p-KR349520.1 transposon, including IS1251 elements. The operon structure of two representative isolates, CriePir1877 with truncated *vanYZ* and CrirPir1878 with full *vanXYZ*, is shown in Figure 3.

In order to find possible interrelation between the presence of truncated *van* operon and other characteristics of clinical isolates, we compared CriePir337, CriePir1877, and SCPM-O-B-8944 with the set of 21 Korean *vanYZ*-reduced clinical isolates from a large-scale *E. faecium* study [19]. However, the cgMLST profiles of Korean isolates were strikingly different (by 300–600 alleles) from any of the first three samples. Even the MLST profiles were different, with only four Korean isolates belonging to ST80^Hom^ (yet assigned to dissimilar or undetermined ST^Bez^), while a total of seven different STs^Hom^ were observed among 21 samples. The complete cgMLST profiles, as well as STs, of all reference isolates are presented in Appendix A.

### 2.3. Plasmid Analysis

The isolates under study were predicted to possess from four (CriePt1087) to ten (CriePt972) plasmid replicons, and the total number of distinct replicons in the set was 18. The most widespread replicon families included rep18, rep11, and repUS15. The genes forming the *van* operon were predicted to be located on plasmids for all isolates.

Hybrid long- and short-read assemblies of CriePir1877, 1879, and 1879 allowed for the obtaining of more precise plasmid structures. The number of plasmids in these isolates was equal to five. All isolates contained a megaplasmid of repUS15 type, which was previously found to be abundant in *Enterococcus* spp. [20]. However, in CriePir1877, this plasmid was larger (282 k) and contained two aminoglycoside resistance genes, while the plasmids of CriePir1878 and 1879 were both about 220 k in size and did not contain any resistance determinants.

The truncated *vanA* operon of CriePir1877 was located on a relatively small 20 k plasmid of rep17 type, which was previously revealed in *E. faecalis* from poultry [21], while the operons of CriePir1878 and 1879 were located on 120 k plasmids, which were very similar to previously described linear pBP5067_P1 from a clinical Indian *E. faecium* isolate (Genbank acc. CP059807.1). On the latter plasmids, no typical relaxase genes were revealed, and no similarity was found with any of the *rep* genes from public databases, which was also the case for the reference plasmid denoted above [22]. Three other plasmids of these three isolates were smaller than 20 k, belonged to the rep11 and rep18 families, and did not contain any resistance or virulence determinants.

### 2.4. Virulence Factors and CRISPR-Cas Systems

The presence of virulence determinants is shown in Figure 4. It is easy to see that the diversity of virulence factors in the *E. faecium* isolates studied was rather low. All isolates possessed the collagen-binding adhesin (*acm*) gene, and almost all included *ecbA/fss3* and *sgrA,* involved in biofilm development [23]. The second collagen adhesin gene *scm*, which is also supposed to be involved in biofilm formation, was present only in CriePir271, the isolate possessing the highest number of virulence factors. At the same time, CriePir269 included only one factor. The virulence genes were predicted to be located on chromosomes for all isolates under study.

Neither CRISPR-Cas systems nor CRISPR arrays with a high evidence level were detected in the *E. faecium* isolates studied. This finding corresponds to previously reported low levels of CRISPR-Cas presence in VREfm isolates [24].

## 3. Discussion

Two major clades of *E. faecium* are currently distinguished, namely, clade A, associated with hospital-acquired infections, and clade B, associated with community-acquired infections [7,25]. Clade A, in turn, divides into the A1 clade with clinical isolates [7] and several other clades (e.g., A2) that include animal and sporadic human infection isolates [7,26]. The clonal complex CC17^Hom^ is a widespread part within the A1 clade and is distributed globally [10]. Multiple surveillance studies have demonstrated the global predominance of the hospital-adapted CC17^Hom^ among clinical *E. faecium* isolates. This lineage has been consistently reported as being predominant in Europe [10], China [27], South Korea [28], and South America [29].

Interestingly, the most common vancomycin-resistant phenotype is *vanA*, while in some countries, in particular Germany and Australia, *vanB* is prevalent [3]. In addition, the *vanM* phenotype was previously revealed in some areas of China [30]. Among numerous STs associated with MDR from CC17^Hom^, ST412^Hom^ was quite common in Brazil, Colombia, Ecuador, Peru, and Venezuela [10,31], while ST78^Hom^ was often revealed in China [32,33]. In Cuba, ST656^Hom^ and ST262^Hom^ occurred, while ST17^Hom^, ST203^Hom^, and ST601^Hom^ were also found in Malaysia, and ST414^Hom^, ST17^Hom^, and ST18^Hom^ were distributed in Taiwan [10]. In Russia, ST17^Hom^, ST78^Hom^, and ST80^Hom^ with *vanA* or *vanB* phenotypes were prevalent among VREfm [34].

In this work, the genomes of ten clinical vancomycin-resistant isolates were analyzed. All isolates in our study were assigned to CC17^Hom^, and all except two belonged to ST80^Hom^. The set included three almost identical isolates, CriePir271–273, another group including CriePt972, CriePt1087, CriePir1878, and CriePir1879 with a slightly different genomic resistance determinant profile, and three single isolates, including CriePir269, 373, and 1877, which were different from other groups in terms of their resistance gene content and cgMLST profiles.

Thus, the isolate classification based on an older MLST^Hom^ scheme did not reflect this diversity, but the newer one, MLST^Bez^, provided much better results since it assigned the first and the second group to different STs^Bez^, yet the same CC152^Bez^, while the singletons were assigned to three different STs^Bez^, including novel ST1523^Bez^ (CriePir1877). At the same time, cgMLST analysis showed that CriePir1878 and 1879 (both—ST144^Bez^) were closer to Pt1087 (ST145^Bez^) than to other members of ST144^Bez^, which is not what is usually expected from ST definitions.

We can conclude that in our case, the MLST^Bez^ was more suitable for distinguishing rather distant isolates, e.g., for the purpose of outbreak investigations, which is in agreement with previous findings [15], but could lead to some misclassifications when considering more similar samples. On the other hand, cgMLST was shown to be more informative than MLST when studying the population structure of VREfm [35]. This was additionally confirmed by our results since the isolates within CC7^Bez^ had a significantly smaller number of differences in their cgMLST alleles with each other than with CriePir269 and 373 assigned to CC152^Bez^ (less than 300 vs. more than 400), but CriePir269 and 373, which were assigned to different STs^Bez^ but the same CC^Bez^, also had a large number of differences in between (namely, 352).

The virulence factor repertoire in our isolates was scarce, with the number of factors ranging from one to four. All isolates included the *acm* gene, which was shown to have a higher incidence within clinical CC17^Hom^ isolates and was even considered as a factor determining the success of *E. faecium* as a nosocomial pathogen [36]. Several biofilm formation factors were revealed variably (*ecbA/fss3* and *sgrA* in seven and nine isolates, respectively, and *scm* was found in only one isolate). However, the virulence factors found to be prevalent in strong biofilm-forming strains (*agg*, *fsrABC*, *gelE*, and *sprE* [37]) were not revealed in the isolates under study, thus demonstrating their low biofilm-formation potential. These results correspond to the recent report, which showed a rather low presence of virulence factors in clinical *E. faecium* isolates in comparison to *E. faecalis* [38].

All VREfm isolates in this study were resistant to key antimicrobial drugs usually used for the treatment of infections caused by *E. faecium*, namely, ampicillin, gentamicin, and vancomycin, which was previously revealed for the members of CC17^Hom^, and ST80^Hom^ in particular [39]. Additional resistance to fluoroquinolones and teicoplanin highlights the necessity of continuous surveillance of such isolates. At the same time, all isolates were susceptible to linezolid, the current rate of resistance to which was estimated to be about 1.2% in the *E. faecium* population [40], and tigecycline, for which the resistance rate was about 3.5% in Europe [41]. These drugs, together with daptomycin, remain to serve as the last resort antibiotics for VREfm [41,42]. The genomic resistance profile corresponded well with the phenotypic testing results, and all isolates were additionally found to carry macrolide (*ermB* or *msrC*) and tetracycline (*tet(L)* and *tet(M)*) resistance genes, which were often encountered in clinical strains [43].

The location of widespread AMR genes, including *aac(6′)-II*, *efmA*, *msrC*, and *tet(L,M)* but excluding the *vanA* operon, was predicted to be on a chromosome, while rarely found genes were predicted to be of plasmid origin, which corresponds to recent findings [32]. Application of several bioinformatics tools allowed the reconstruction of full *vanA* operons, although the precise structure of plasmids carrying them was obtained for CriePir1877, 1878, and 1879 only.

All the genomes possessed a *vanA* resistance genotype, which was a prevailing type worldwide during recent decades and still holds its position currently [19,44,45]. Resistance gene clusters were located on a Tn1546 transposon, which is a characteristic of *vanA* genotypes and was previously shown to mediate vancomycin resistance transfer via plasmids in *E. faecium* [19,32,46]. This fact provides insights into possible *vanHAX* acquisition mechanisms within the VREfm under study. The *vanA* operons of eight isolates comprised the full-length clusters of seven genes, while the operons of the isolates CriePir337 and CriePir1877 had a truncated vancomycin resistance gene cassette lacking *vanY* and *vanZ* genes. It should be noted that both isolates retained the resistance to vancomycin and teicoplanin despite the operon truncation.

The VREfm isolates from the current study were not close to any reference isolates, with the exception of CriePir337, which possessed a striking similarity with a clinical isolate SCPM-O-B-8944 (Genbank acc. GCA_020405705.1), obtained in another Russian hospital from the human digestive system of a single patient in 2020. Similar to CriePir337, SCPM-O-B-8944 had a reduced *vanA* operon, consisting of five genes only (*vanRSHAX*). This isolate had the same MLST type ST909^Bez^/ST80^Hom^ and a cgMLST profile very similar to that of CriePir337. SCPM-O-B-8944 was a member of the group, also including SCPM-O-B-8940, SCPM-O-B-8941, and SCPM-O-B-8942, obtained from the same source and during the same time interval, which also had cgMLST profiles very similar to each other (the number of allelic differences within the group was less than 16, thus allowing their assignment to a single clone [47]). Worth noting, other members of this group included the intact *vanA* operon, which confirmed that such a deletion was not an assembly artifact and served as an independent reliability confirmation of our finding of the same event in CriePir337. Unfortunately, no phenotypic resistance and other metadata were provided for the above reference isolates, thus making it impossible to perform a comprehensive comparison.

Another set of clinical isolates with a truncated *vanA* operon lacking *vanYZ* genes was revealed in South Korea during a wide-scale research of VREfm blood isolates carrying *Tn1546*-type transposons [19]. In this study involving 308 VREfm isolates, 21 had *vanA* phenotypes with truncated *vanA* gene clusters lacking *vanY* and *vanZ* genes. However, these isolates had an additional copy of Tn1546, harboring either a full-length *vanA* operon or gene cassette ended with a truncated *vanY* gene. The Korean *vanYZ*-reduced isolates, in general, possessed different STs and were neither close by their cgMLST profiles to each other nor to CriePir337 and 1877. Plasmid analysis also did not elucidate the possible characteristic properties of *vanYZ*-reduced clones.

Due to high genome plasticity, enterococci can easily acquire various genetic resistance determinants that allow bacteria to adapt to diverse environments [3]. However, the fitness cost of resistance gene acquisition might be high regardless of whether the genes are located on a plasmid or a chromosome. Kim et al. [19] observed that blood isolates of *E. faecium* harboring the *Tn1546* transposon carrying a *vanA* operon grew significantly slower in antimicrobial-free medium than those lacking the vancomycin resistance gene cluster. This observation allowed the researchers to suggest that it is rather costly for bacteria to maintain the *vanA* operon if the antimicrobial drug is absent in the environment [3,19]. The detection of isolates harboring similarly truncated *vanA* operons in distinct geographic locations—including another Russian hospital and healthcare facilities in South Korea—suggests that these clones may possess a selective advantage facilitating their dissemination under specific environmental or antimicrobial pressures. This observation warrants further investigation into the potential epidemiological fitness conferred by the reduced *vanA* gene cluster.

Furthermore, Kim et al. [19] suggested that the absence of vancomycin in the environment could lead to the emergence of vancomycin-variable isolates that are susceptible to vancomycin but retain a structurally impaired *vanA* operon that is completely inactive. The variable clones revealed in this study had the same growth rates as vancomycin-negative isolates without *vanA* gene clusters, which points to another mechanism of decreasing the fitness burden for VREs than just removing the *vanA* operon from the genome. Consequently, reducing the number of glycopeptides in the environment, in particular, their justified prescription, can significantly reduce the number of vancomycin-resistant isolates and prevent their dissemination.

Comprehensive WGS-based studies of various *van* operon structures, including their reduced and ‘silent’ versions, will advance our understanding of vancomycin and teicoplanin resistance mechanisms in *E. faecium*. Such investigations may facilitate the development of targeted strategies to curb resistance dissemination, including optimized antibiotic stewardship programs and novel molecular interventions such as CRISPR-Cas-based antimicrobials or phage therapy [48,49].

## 4. Materials and Methods

### 4.1. Sample Collection, Antibiotic Susceptibility Testing, and DNA Isolation

In total, 47 isolates of *E. faecium* were obtained from three participating hospitals (referenced in Table 1 as A, B, and C) during the period of 2019–2024 as a part of routine microbiological monitoring, and ten vancomycin-resistant isolates were selected for the study. Specimen details are provided in the Section 2.

The isolation of pure bacterial cultures was performed by seeding on solid nutrient media (Endo Agar) with subsequent species identification based on morphological, biochemical, and antigenic properties. The species identification was also confirmed by time-of-flight mass spectrometry (MALDI-TOF MS) using the VITEK MS system (bioMerieux, Marcy-l’Étoile, France).

Antimicrobial susceptibility profiling was performed using two complementary approaches: conventional Kirby–Bauer disk diffusion methodology employing Mueller–Hinton agar (bioMerieux, Marcy-l’Étoile, France) with antibiotic-impregnated disks (BioRad, Marnes-la-Coquette, France), and automated minimum inhibitory concentration determination using the VITEK2 Compact 30 system (bioMerieux, Marcy-l’Étoile, France). The antimicrobial panel included representatives from major drug classes to assess resistance patterns, with particular attention to clinically relevant agents. The panel included the following antimicrobial drugs: ampicillin, gentamicin, levofloxacin, linezolid, streptomycin, teicoplanin, tigecycline, and vancomycin. The EUCAST clinical breakpoints version 12 was used to interpret the results obtained (https://www.eucast.org/clinical_breakpoints/, accessed on 20 December 2022).

Genomic DNA was isolated using a DNeasy Blood and Tissue kit (Qiagen, Hilden, Germany) and used for the paired-end library preparation with Nextera™ DNASamplePrepKit (Illumina^®^, San Diego, CA, USA). The WGS of ten isolates was performed on Illumina^®^ HiSeq1500 and NextSeq2000 platforms (Illumina^®^, San Diego, CA, USA). Three isolates were additionally sequenced on an Oxford Nanopore MinION sequencing system (Oxford Nanopore Technologies, Oxford, UK). The libraries were prepared according to the manufacturer’s protocols with a Rapid Barcoding Sequencing kit SQK-RBK004 (Oxford Nanopore Technologies, Oxford, UK) and were sequenced on a FLO-MIN106 R9.4 flow cell. The isolates were selected for long-read sequencing based on their genomic properties (possessing a dominating ST or including a truncated *vanA* operon) and the quality of the isolated DNA.

### 4.2. Genome Assembly, Data Processing, and Annotation

Short-read assemblies were obtained using SPAdes versions 3.13.0, 3.15.2, and 3.15.4. Hybrid short- and long-read assemblies were obtained using Unicycler version 0.5.0 (normal mode) [50]. All assemblies were uploaded to NCBI Genbank under the project number PRJNA1151717.

Multilocus sequence typing (MLST) and the corresponding clonal complex assignment for the isolates studied was made using two schemes, namely, the original one developed by Homan et al. [14] (including seven genes, namely *adk*, *atpA*, *ddl*, *gdh*, *gyd*, *pstS*, and *purK*), and the novel one developed in 2023 by Bezdicek et al. (including eight genes, namely *copA*, *dnaE*, *HP2027*, *mdlA*, *narB*, *pbp2B*, *rpoD*, and *uvrA*), which was intended to provide better resolution [15]. For both schemes, PubMLST was used to obtain sequence type (ST) and clonal complex (CC) definitions (https://pubmlst.org/bigsdb?db=pubmlst_efaecium_seqdef, accessed on 10 June 2025). In order to avoid confusion, STs and CCs are marked here as ST(CC)^Hom^ for the Homan scheme and ST(CC)^Bez^ for the novel Bezdicek scheme, respectively.

The cgMLST profiles for *E. faecium* isolates were obtained with MentaList [51] (https://github.com/WGS-TB/MentaLiST, version 0.2.4, default parameters, accessed on 11 June 2025) using the scheme containing 1423 loci developed by de Been et al. [52] (https://www.cgmlst.org/ncs/schema/Efaecium4420/, accessed on 11 June 2025). The minimum spanning trees were built using PHYLOViz online (http://online.phyloviz.net, accessed on 15 June 2025).

The Resfinder 4.6.0 [53] (http://genepi.food.dtu.dk/resfinder, accessed on 10 June 2025) and CARD [54] (https://card.mcmaster.ca/, accessed on 11 June 2025) databases were used to predict antimicrobial resistance genes. Resfinder predicts only the HAX operon as a whole, while CARD makes a more detailed prediction for the positions of the genes composing this operon. The PointFinder database of Resfinder was used to predict point mutations conferring AMR to the isolates, in particular, to ampicillin and ciprofloxacin/levofloxacin.

Searching for virulence factors in the *E. faecium* genomes was performed using VFDB [55] with default parameters (http://www.mgc.ac.cn/VFs/main.htm, accessed on 11 June 2025).

We used mlplasmids [56] and Plasmer [57] with default parameters in order to predict which contigs in the assemblies were likely to belong to plasmids, and only the contigs predicted as plasmidic by both programs were considered as being of plasmid origin. Then we used the mob_recon program version 3.1.9 from the MOB-suite package [58] with default parameters to cluster the contigs of plasmidic origin into separate groups likely corresponding to plasmids. We selected the cluster containing the contigs carrying the *vanA* operon in order to remap and reassemble raw reads, which were mapped to these contigs. Using this procedure, we successfully reconstructed the whole *vanA* operons for the isolates. The procedure described above was used for short-read-based assemblies.

Searching for transposons was performed using the TnCentral web server [59] (https://tncentral.ncc.unesp.br/blast/, accessed on 17 June 2025) with default parameters.

The detection of CRISPR arrays and CRISPR-Cas systems was made with CRISPRCasFinder [60] version 4.2.20 with the following parameters: ‘-cas –fast –rcfowce -ccvRep -vicinity 1100 -useProkka’.

We used the pipeline developed earlier by us for seamless integration of the data obtained from the sources above and visualization of the results [61,62].

## 5. Conclusions

In this study, we performed whole-genome sequencing (WGS) and comprehensive characterization of 10 vancomycin-resistant *Enterococcus faecium* (VREfm) clinical isolates to investigate the genetic determinants of vancomycin and teicoplanin resistance, with a particular focus on the structure of the *vanA* operon. Phylogenetic analysis, based on isolate typing and comparison with reference strains from public databases, highlighted the advantages and limitations of existing multilocus sequence typing (MLST) schemes, confirming that the core genome MLST (cgMLST) offers superior resolution for molecular epidemiology.

Two isolates harbored a truncated *vanA* operon lacking *vanY* and *vanZ* genes, and the plasmid carrying this operon was successfully reconstructed. Comparative genomic analysis of these isolates and publicly available strains with similar truncations did not reveal significant genetic relatedness, suggesting diverse evolutionary origins and warranting further investigation. Additionally, our study identified a variant of the *vanA* operon, which may represent an emerging resistance determinant.

These findings provide crucial baseline data for the epidemiological surveillance of VREfm infections in Russia and contribute to the global understanding of vancomycin resistance mechanisms in *E. faecium*. Further studies are needed to explore the clinical implications of the truncated *vanA* operon and the dissemination patterns of the identified variants.

## Figures and Tables

**Figure 1 antibiotics-14-00858-f001:**
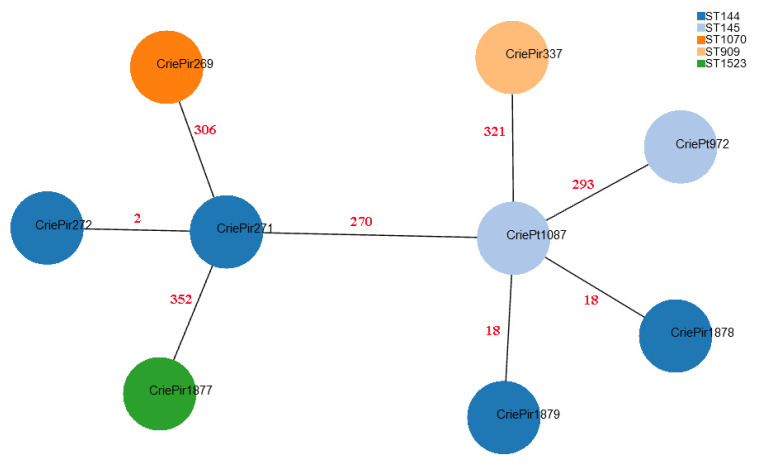
Minimum spanning tree based on cgMLST profiles of clinical *E. faecium* isolates. CriePir273 is not shown since its cgMLST profile was identical to that of CriePir271. The nodes are colored based on ST^Bez^ according to the legend provided. Red numbers indicate the number of cgMLST allele differences between the corresponding pairs of the isolates.

**Figure 2 antibiotics-14-00858-f002:**
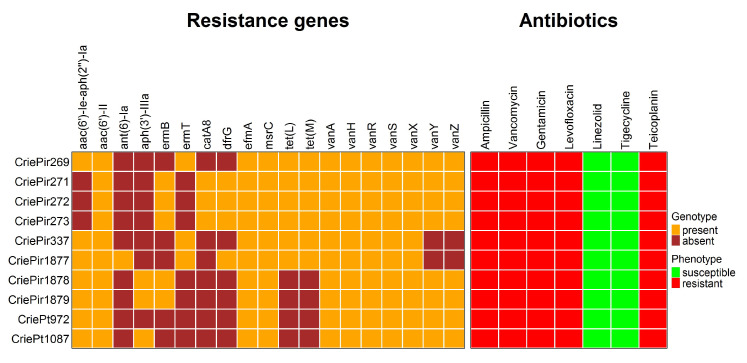
Genomic and phenotypic antimicrobial resistance of 10 clinical *E. faecium* isolates under study. Note the absence of *vanY* and *vanZ* genes in the isolates CriePir337 and 1877 carrying truncated *vanA* operons.

**Figure 3 antibiotics-14-00858-f003:**
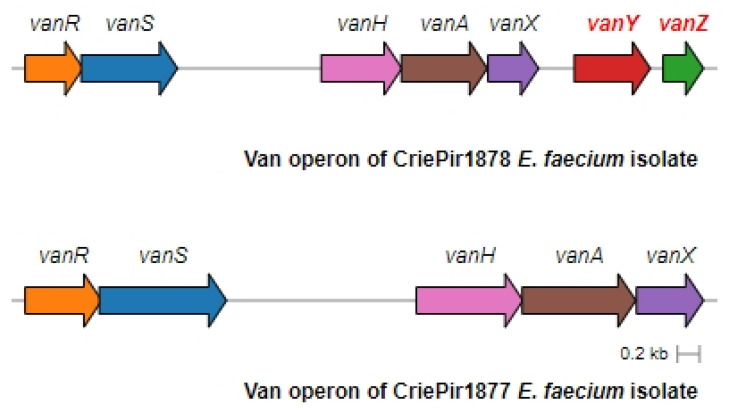
Full and truncated *vanA* operon structures for the representative *E. faecium* isolates. CriePir1878 includes *vanY* and *vanZ* genes missing in CriePir1877.

**Figure 4 antibiotics-14-00858-f004:**
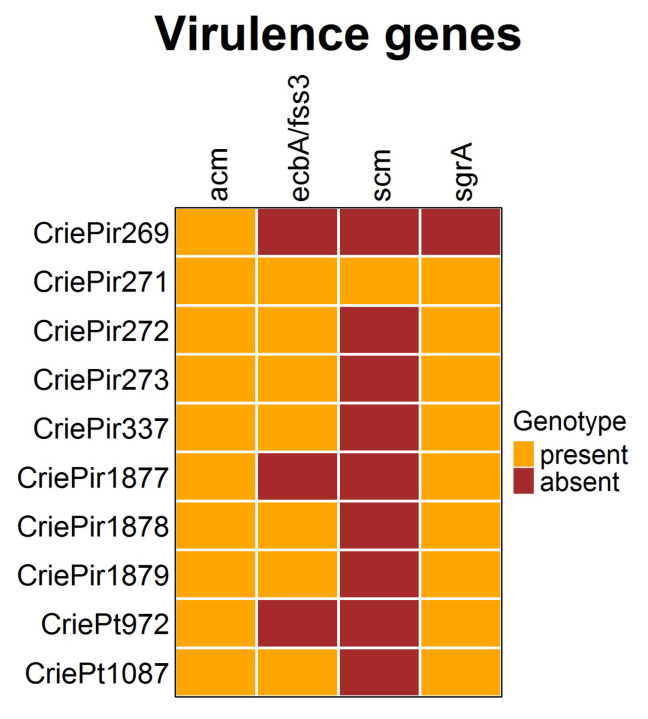
Virulence factors of clinical *E. faecium* isolates. The diversity was rather low.

**Table 1 antibiotics-14-00858-t001:** Typing results for clinical VRE isolates.

Isolate ID	ST^Hom^	CC^Hom^	ST^Bez^	CC^Bez^	Specimen	Hospital	Year of Isolation
CriePir269	ST552	CC17	ST1070	CC7	urine	A	2019
CriePir271	ST80	CC17	ST144	CC152	feces	A	2019
CriePir272	ST80	CC17	ST144	CC152	blood	A	2019
CriePir273	ST80	CC17	ST144	CC152	urine	A	2019
CriePir337	ST80	CC17	ST909	CC7	urine	A	2020
CriePir1877 *	ST18	CC17	ST1523	CC128	apostem	A	2024
CriePir1878 *	ST80	CC17	ST144	CC152	urine	A	2024
CriePir1879 *	ST80	CC17	ST144	CC152	wound	A	2024
CriePt972	ST80	CC17	ST145	CC152	urine	B	2022
CriePt1087	ST80	CC17	ST145	CC152	urine	C	2022

* indicates the isolates sequenced on MinION (long reads).

## Data Availability

Genome assemblies for the isolates discussed in the manuscript were uploaded to NCBI Genbank under the project number PRJNA1151717.

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
