# Peer review of "Genomic Epidemiology of Vancomycin-Resistant Enterococcus faecium Isolates with Full and Truncated vanA Operons from Russian Hospitals"

_antibiotics, 2025, doi:10.3390/antibiotics14090858_

Round 1

Reviewer 1 Report

Comments and Suggestions for Authors

I would like to congratulate the authors for the realized work. My main concern about the manuscript is the discussion part. The studied strains were isolated over a five year period (between 2019 and 2024). the sidcussion demonstrated genetic heterogeneity among clinical VREfm isolates, but did not explore whether these differences reflected temporal evolution. I believe that this temporal range is sufficiently long to investigate meaningful genomic trends in Enterococcus faecium, particularly in the context of resistance gene acquisition or loss, clonal expansion or replacement, emergence of novel variants (truncated vanA operons), shifts in distribution and changes in virulence gene profiles ... 

Lign 2 : Enterococcus faecium should be in Italic, 

Lign 6 : Central instead of 1Central (delete the 1), 

Some keywords are already present in the abstract text, please change for a bibliographic interest (MLST scheme; truncated vanA operon). Please reorder keywords A-Z. 

Reference [1], can you use a more recent one for taxonomy, 

Lign 35 : "These bacteria are able to survive under severe environment conditions including high temperatures and saline concentrations." any specific values (Max tempertures/Salinity) ?  

Lign 49 : "in particular, since 1960s, strains with resistance to streptogramines, aminoglycosides, macrolides, lincosamides, tetracyclines, glycopeptides and to cyclic lipopeptides were discovered" become : "Since 1960s, strains ...",

Ling 51 : "Since 1990s," become "from the 1990s" to avoid repeated "Since", 

Lign 92 and 94 : Homan et al. and Bezdicek et al. are first mentionned here, but not as references in the manuscript. I believe they should  [?], [?]. 

Figures : Please enhance all figures' captions. A figure should be easy to read, even when it stands alone. 

Author Response

We thank the reviewers for their work on the manuscript and the comments leading to its improvement.

Reviewer 1.

I would like to congratulate the authors for the realized work. My main concern about the manuscript is the discussion part. The studied strains were isolated over a five year period (between 2019 and 2024). the sidcussion demonstrated genetic heterogeneity among clinical VREfm isolates, but did not explore whether these differences reflected temporal evolution. I believe that this temporal range is sufficiently long to investigate meaningful genomic trends in Enterococcus faecium, particularly in the context of resistance gene acquisition or loss, clonal expansion or replacement, emergence of novel variants (truncated vanA operons), shifts in distribution and changes in virulence gene profiles ... 

We agree that the temporal range was long enough to study the evolutionary genomic differences, but the number of the isolates was not. Since the evolution can be better traced within particular STs, not between them, the groups of similar STs should be studied for these purposes, and the largest group contained just 8 (or just 5 according to another scheme) isolates, which was too low to study the genomic trends. We hope to perform such a study in future when more isolates will become available.

Lign 2 : Enterococcus faecium should be in Italic, 

Fixed

Lign 6 : Central instead of 1Central (delete the 1), 

Done

Some keywords are already present in the abstract text, please change for a bibliographic interest (MLST scheme; truncated vanA operon). Please reorder keywords A-Z. 

Done

Reference [1], can you use a more recent one for taxonomy, 

Reference updated

Lign 35 : "These bacteria are able to survive under severe environment conditions including high temperatures and saline concentrations." any specific values (Max tempertures/Salinity) ?  

The corresponding values were added

Lign 49 : "in particular, since 1960s, strains with resistance to streptogramines, aminoglycosides, macrolides, lincosamides, tetracyclines, glycopeptides and to cyclic lipopeptides were discovered" become : "Since 1960s, strains ...",

Fixed as suggested

Ling 51 : "Since 1990s," become "from the 1990s" to avoid repeated "Since", 

Fixed as suggested

Lign 92 and 94 : Homan et al. and Bezdicek et al. are first mentionned here, but not as references in the manuscript. I believe they should  [?], [?]. 

We added the references at first appearance of the schemes

Figures : Please enhance all figures' captions. A figure should be easy to read, even when it stands alone. 

We added the captions as requested

Reviewer 2 Report

Comments and Suggestions for Authors

Please go through the whole manuscript carefully and correct it according to the comments and suggestions.

Author Response

We made the changes according to the comments provided in pdf file. However, we decided to leave the highlighted paragraphs and not move them to Discussion section since they, in fact, only show the results and the mere fact of having references inside them does not substantiate such a move.

We also did not add references to Vitek and Illumina equipment – the manufactures and their location was already described in the manuscript.

Reviewer 3 Report

Comments and Suggestions for Authors

This study characterized ten clinical vancomycin resistant E. faecium isolates from Russia. The study is well designed. The study underscore the importance of genomic surveillance to track resistant strains. Core genome MLST has provided better resolution and has resolved strain relationship and WGS identified presence of van operon located on plasmids highlighting horizontal gene transfer in resistance dissemination. 

I have following suggestions for authors

  1. Figures can be improved. Figure 1 and 3 appear stretched and blurred.
  2. In line 162, author refer to figure but does not include its corresponding number.
  3. There is a discrepancy in results and discussion, In line 92, it is stated that  "According to the older MLST scheme by Homan et al. (MLSTHom), all isolates were  assigned to CC17Hom, and eight of them belonged to ST80Hom. " However in  line 229, they mentioned "All isolates in our study were assigned to CC17Hom, and all except one belonged to ST80Hom."
  4. In methods, please provide rational for selecting only three isolates for long read sequencing.
Comments on the Quality of English Language

The quality looks fine but there is room for improvement. 

Author Response

This study characterized ten clinical vancomycin resistant E. faecium isolates from Russia. The study is well designed. The study underscore the importance of genomic surveillance to track resistant strains. Core genome MLST has provided better resolution and has resolved strain relationship and WGS identified presence of van operon located on plasmids highlighting horizontal gene transfer in resistance dissemination. 

I have following suggestions for authors

  1. Figures can be improved. Figure 1 and 3 appear stretched and blurred.

Figures are now provided in better resolution

  1. In line 162, author refer to figure but does not include its corresponding number.

Fixed (fig. 3 was meant)

  1. There is a discrepancy in results and discussion, In line 92, it is stated that  "According to the older MLST scheme by Homan et al. (MLSTHom), all isolates were  assigned to CC17Hom, and eight of them belonged to ST80Hom. " However in  line 229, they mentioned "All isolates in our study were assigned to CC17Hom, and all except one belonged to ST80Hom."

Fixed to ‘all except two belonged’ in the latter case

  1. In methods, please provide rational for selecting only three isolates for long read sequencing.

The statement was added to section 4.1